# First-in-human phase Ia study of the PI3Kα inhibitor CYH33 in patients with solid tumors

Xiao-Li Wei[1,6], Fu-Rong Liu[2,6], Ji-Hong Liu[3,6], Hong-Yun Zhao[2], Yang Zhang[2], Zhi-Qiang Wang[1], Miao-Zhen Qiu [1], Fei Xu [1], Qiu-Qiong Yu[4], Yi-Wu Du[4], Yan-Xia Shi[1], De-Sheng Wang [1], Feng-Hua Wang[1] & Rui-Hua Xu [1,5] ✉

PIK3CA mutations are highly prevalent in solid tumors. Targeting phosphatidylinositol 3-kinase α is therefore an attractive strategy for treating cancers harboring PIK3CA mutations. Here, we report the results from a phase Ia, open label, dose-escalation and -expansion study (NCT03544905) of CYH33, a highly selective PI3Kα inhibitor, in advanced solid tumors. The primary outcomes were the safety, tolerability, maximum tolerated dose (MTD) and recommended phase 2 dose (RP2D) of CYH33. The secondary outcomes included evaluation of pharmacokinetics, preliminary efficacy and changes in pharmacodynamic biomarkers in response to CYH33 treatment. The exploratory outcome was the relationship between the efficacy of CYH33 treatment and tumor biomarker status, including PIK3CA mutations. A total of 51 patients (19 in the dose escalation stage and 32 in the dose expansion stage) including 36 (70.6%) patients (4 in the dose escalation stage and 32 in the dose expansion stage) with PIK3CA mutations received CYH33 1–60 mg. The MTD of CYH33 was 40 mg once daily, which was also selected as the RP2D. The most common grade 3/4 treatment-related adverse events were hyperglycemia, rash, platelet count decreased, peripheral edema, and fatigue. Forty-two out of 51 patients were evaluable for response, the confirmed objective response rate was 11.9% (5/42). Among 36 patients harboring PIK3CA mutations, 28 patients were evaluable for response, the confirmed objective response rate was 14.3% (4/28). In conclusion, CYH33 exhibits a manageable safety profile and preliminary anti-tumor efficacy in solid tumors harboring PIK3CA mutations.

PIK3CA, which encodes the p110α catalytic subunit of phosphatidylinositol 3-kinase α (PI3Kα)[1], is one of the most frequently-mutated cancer-associated genes[2], which is often implicated in tumorigenesis[3] and drug resistance[4,5]. For instance, the frequency of PIK3CA mutations is estimated to be 20–46% in ovarian clear cell carcinoma[6,7], 12–40% in endometrioid carcinoma[7], 13–28% in colorectal cancer, 14–23% in cervical cancer[2,8], and up to 40% in hormone receptor positive/human epidermal growth factor receptor 2 (HER2) negative (HR+/HER2−)

[1]Department of Medical Oncology, State Key Laboratory of Oncology in South China, Collaborative Innovation Center for Cancer Medicine, Sun Yat-sen University Cancer Center, Sun Yat-sen University, Guangzhou 510060, China. [2]Department of Clinical Research, State Key Laboratory of Oncology in South China, Collaborative Innovation Center for Cancer Medicine, Sun Yat-sen University Cancer Center, Sun Yat-sen University, Guangzhou 510060, China. [3]Department of Gynecologic Oncology, State Key Laboratory of Oncology in South China, Collaborative Innovation Center for Cancer Medicine, Sun Yat-sen University Cancer Center, Sun Yat-sen University, Guangzhou 510060, China. [4]Haihe Biopharma Co., Ltd, Shanghai 201203, China. [5]Precision Diagnosis and Treatment for Gastrointestinal Cancer, Chinese Academy of Medical Sciences, Guangzhou 510060, China. [6]These authors contributed equally: Xiao-Li Wei, Fu-Rong Liu, Ji-Hong Liu. ✉e-mail: xurh@sysucc.org.cn

breast cancer[9,10]. Furthermore, three hotspots (H1047R, E542K, and E545K)[1] constitute around 80% of all somatic driver mutations affecting the *PIK3CA* gene[11].

To date, several PI3K inhibitors have been approved for use in clinical trials[12,13]. Among them, alpelisib is the first and only US FDA-approved selective PI3Kα inhibitor[14]. Results from the phase III SOLAR-1 study showed a 7.9-month improvement in the median overall survival of patients with advanced *PIK3CA*-mutated HR+/HER2− breast cancer when alpelisib was used in combination with fulvestrant, an estrogen receptor antagonist, compared to fulvestrant plus placebo[15]. Another PI3Kα inhibitor, GDC-0077[16], is currently undergoing late-stage clinical investigation. However, the number and structural diversity of selective PI3Kα inhibitors currently available for use in clinical trials is limited. Furthermore, the majority of PI3Kα inhibitors have been developed to treat breast cancer in combination with endocrine therapy. In view of the high prevalence of *PIK3CA* mutations in all cancers, it remains critical to explore the role of PI3Kα inhibitors in a wider variety of tumor types. The discovery of new PI3Kα inhibitors and their application in a variety of disease settings will be of the utmost benefit to patients.

CYH33 is an oral, highly selective PI3Kα inhibitor with an active metabolite, I27. It has been demonstrated that the affinities of CYH33 and I27 for PI3Kα and its mutants ($IC_{50}$ range 5–20 nM) are greater than other isoforms of PI3K (e.g., PI3Kβ, PI3Kδ, and PI3Kγ; $IC_{50}$ range 43.0–611.2 nM)[17]. CYH33 has also been shown to significantly inhibit the proliferation of human breast cancer cells in vitro[18,19]. Furthermore, when used in combination with radiation or a cyclin-dependent kinase (CDK)4/6 inhibitor, CYH33 synergistically inhibited the proliferation of esophageal squamous cell carcinoma[20] and *KRAS*-mutated non-small cell lung cancer cell lines[21].

In this work, we report the results from the first-in-human phase Ia trial of CYH33 in advanced solid tumors. Here, we show that CYH33 exhibits a manageable safety profile and preliminary anti-tumor efficacy in solid tumors harboring *PIK3CA* mutations.

## Results
### Patient characteristics and disposition
From July 13, 2018 to March 29, 2021, a total of 51 patients (median age, 54 years) were enrolled in the phase Ia study and received CYH33 (Table 1). Of these patients, 19 patients were recruited to the dose-escalation stage, with 15 (78.9%) patients having unknown *PIK3CA* mutation status and 4 (21.1%) patients harboring *PIK3CA* mutations. Thirty-two (100%) patients harboring *PIK3CA* mutations determined via local laboratory testing were included in the dose-expansion stage, 4 patients of them in the 20 mg group, 12 in the 30 mg group, and 16 in the 40 mg dose group (Fig. 1). At baseline, a total of 18 (35.3%) patients had an Eastern Cooperative Oncology Group (ECOG) score of 0 and 33 (64.7%) patients had a score of 1. The primary tumor types were categorized as colorectal cancer (n = 10, 19.6%), breast cancer (n = 6, 11.8%), ovarian cancer (n = 6, 11.8%), cervical cancer (n = 6, 11.8%), endometrial cancer (n = 5, 9.8%), nasopharyngeal carcinoma (n = 5, 9.8%), urinary system neoplasm (n = 4, 7.8%), and other cancer types (n = 9, 17.6%). Of these 51 patients, 19 (37.3%) had previously received first-line systemic therapy, 12 (23.5%) had previously received second-line systemic therapy, and 20 (39.2%) had been heavily treated with at least 3 lines of systemic therapy (Table 1).

### Dose escalation and dose-limiting toxicity (DLT)
In the dose-escalation stage, 19 patients were treated once daily with the following doses of CYH33: 1 mg (n = 1), 5 mg (n = 1), 10 mg (n = 2), 20 mg (n = 2), 40 mg (n = 8), or 60 mg (n = 5). The 30 mg dose level was skipped based on a decision by the Safety Monitoring Committee (SMC). Three of the 19 patients experienced a DLT: grade 3 hyperglycemia at the dose of 40 mg (n = 1), grade 3 hyperglycemia and grade 3 nausea at the dose of 60 mg (n = 2). The DLT rate at the 40 mg dose

level was 16.7% (1 out of 6 DLT evaluable patients), which met the requirements of the modified toxicity probability interval 2 (mTPI-2) model[22]. The CYH33 single agent maximum tolerated dose (MTD) was therefore established as a once-daily dose of 40 mg.

### Safety, tolerability and recommended phase 2 dose (RP2D)
All patients from both the dose escalation and dose expansion stages were included in the safety analysis (n = 51). The median exposure to CYH33 was 6.1 weeks (range, 0.4–71.9) at all dose levels and 6.2 weeks (range, 0.4–71.9) for patients receiving 40 mg. The most frequent treatment-related adverse event (TRAE) was hyperglycemia, an on-target toxicity associated with PI3K inhibition[23], which was reported in 90.2% (n = 46) of patients at any grade (Table 2). Other frequent all-grade TRAEs (≥10% of patients) were: decreased appetite (41.2%), nausea (37.3%), weight loss (31.4%), diarrhea (29.4%), vomiting (25.5%), peripheral edema (25.5%), fatigue (17.6%), rash (17.6%), facial edema (15.7%), hyponatremia (11.8%) and mouth ulceration (11.8%). The most frequent grade ≥3 TRAE was hyperglycemia (58.8%). Except for grade 3 peripheral edema in four (7.8%) patients, the incidences of all other grade ≥3 TRAEs were <5% (Table 2). The incidence of hyperglycemia increased in a dose-dependent manner following CYH33 administration, but was effectively managed by appropriate interventions, including dose interruption/reduction and concomitant oral anti-diabetic medications, with or without insulin. In total, 70.6% (n = 36) of patients experienced dose interruptions due to TRAEs (Table 2), with hyperglycemia being the most common TRAE leading to dose interruption (n = 32, 62.7%). Six patients (11.8%) experienced dose reductions due to TRAEs including fatigue (n = 2, 4.0%), peripheral edema, weight loss, mouth ulceration, and prolonged electrocardiogram QT (n = 1, 2.0% for each). Three patients (5.9%) permanently discontinued CYH33 due to DLTs. Six patients (11.8%) experienced treatment-associated serious AEs: nausea, vomiting, abdominal pain, dyspnea, hyperglycemia, and pyrexia (n = 1 each, 2.0% for each). No treatment-related deaths occurred. Therefore, the RP2D was set at 40 mg QD.

**Table 1 | Patient demographics at baseline**

| Characteristic | DL1–4 CYH33, 1–20 mg QD n = 10 | DL5 CYH33, 30 mg QD n = 12 | DL6 CYH33, 40 mg QD n = 24 | All patients CYH33, 1–60 mg QD n = 51 |
|---|---|---|---|---|
| Age (years) median (range) | 55.5 (47, 67) | 54.5 (34, 70) | 49.0 (26, 73) | 54.0 (26, 73) |
| Sex (male/female) | 6/4 | 4/8 | 9/15 | 22/29 |
| ECOG status, n (%) | | | | |
| 0 | 4 (40.0) | 3 (25.0) | 10 (41.7) | 18 (35.3) |
| 1 | 6 (60.0) | 9 (75.0) | 14 (58.3) | 33 (64.7) |
| Number of prior systemic regimens, n (%) | | | | |
| 1 | 2 (20.0) | 7 (58.3) | 10 (41.7) | 19 (37.3) |
| 2 | 2 (20.0) | 2 (16.7) | 6 (25.0) | 12 (23.5) |
| 3 | 2 (20.0) | 0 | 3 (12.5) | 8 (15.7) |
| ≥4 | 4 (40.0) | 3 (25.0) | 5 (20.8) | 12 (23.5) |
| Number of metastatic sites, n (%) | | | | |
| 1 | 0 | 1 (8.3) | 7 (29.2) | 9 (17.6) |
| 2 | 4 (40.0) | 4 (33.3) | 7 (29.2) | 16 (31.4) |
| ≥3 | 6 (60.0) | 7 (58.3) | 10 (41.7) | 26 (51.0) |
| Status of *PIK3CA*, n (%) | | | | |
| Mutant | 4 (40.0) | 12 (100.0) | 17 (70.8) | 36* (70.6) |
| Wild type | 0 | 0 | 0 | 0 |
| Unknown | 6 (60.0) | 0 | 7 (29.2) | 15 (29.4) |

*Three patients in the 60 mg dose group harbored *PIK3CA* mutation status.
*DL* dose level, *QD* once-daily, *ECOG* Eastern Co-operative Oncology Group.

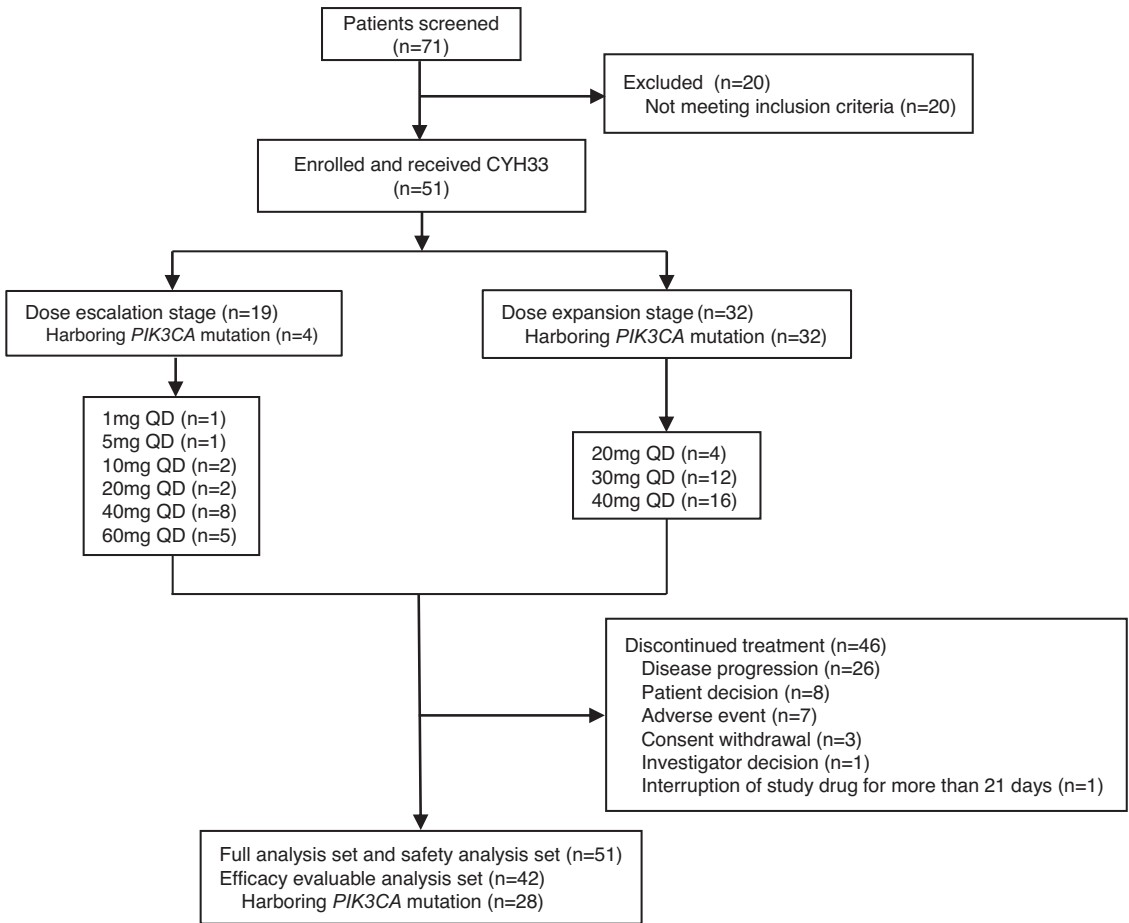

**Fig. 1 | Trial profile.** Diagram indicating participant numbers and disposition through the course of the trial. QD, once daily.

## Pharmacokinetic (PK) and pharmacodynamic (PD) analysis

The PK characteristics of CYH33 and its active metabolite I27 were evaluated after single and multiple dose administrations. PK data were obtained from 37 patients from 7 dose cohorts: 1 mg ($n = 1$), 5 mg ($n = 1$), 10 mg ($n = 2$), 20 mg ($n = 6$), 30 mg ($n = 12$), 40 mg ($n = 13$), and 60 mg ($n = 2$). Intensive and sparse PK sampling was performed. Prior to initiating a continuous daily treatment cycle, the first 5 patients participated in a single dose PK treatment period with a 7-day washout in order to determine the terminal-phase half-life ($t_{1/2}$) of CYH33 and its active metabolite I27. Based on the single dose PK data for CYH33 of the first 5 patients at doses of 1, 5, 10 and 20 mg, the PK and metabolic characteristics of CYH33 and I27 were evaluated. Combined with the pre-specified rules in the protocol and deliberation from the SMC, subsequent patients directly entered the continuous dose phase.

Following oral administration, CYH33 was quickly absorbed with a median time to maximum plasma concentration ($T_{max}$) of 1–4 h in the dose range of 1–60 mg (Fig. 2a, Table 3). The systemic exposure to CYH33 and I27 (maximum plasma concentration [$C_{max}$], area under the plasma concentration-time curve from time 0 to 24 h [$AUC_{0-24h}$]) increased with dose after single and multiple dose administrations (Fig. 2a–d, Table 3 and Supplementary Tables 1–3). The $t_{1/2}$ for CYH33 and I27 was approximately 20.0 h and 27.8 h, respectively, and a steady state was reached by day 8 following daily dosing. The average accumulation of exposure ($AUC_{0-24h}$) at steady state was around 2.9- and 4.3-fold for CYH33 and I27 compared to Cycle 1 Day 1 (C1D1), respectively (Table 3, Supplementary Tables 1–3).

Since glucose metabolism is tightly regulated by PI3Kα[24], fasting blood glucose (FBG) levels were used as a PD marker. A dose-dependent increase in FBG levels was observed on Cycle 1 Day 15 (C1D15), starting at 10 mg CYH33 and peaking at 60 mg (Fig. 2e).

## Radiological efficacy

Of the 51 patients in the study, 42 were included in the final efficacy evaluable analysis set (28 patients harboring *PIK3CA* mutations and 14 with unknown *PIK3CA* mutation status); 9 patients missed their post-baseline assessments due to AEs or voluntary withdrawal. As of the data cut-off date 16th July 2021, half of the 42 evaluable patients experienced a shrinkage of the target lesions compared with baseline. Five patients achieved a confirmed tumor response including 1 complete response and 4 partial responses (PR), among whom 4 patients had *PIK3CA* mutations in the dose-expansion stage and one colorectal cancer patient had unknown *PIK3CA* mutation status in the dose escalation stage. The confirmed objective response rate (ORR; CR + PR) was therefore 11.9% (5/42; 95% confidence interval [CI], 3.98–25.63) and the disease control rate (DCR; CR + PR + stable disease (SD) ≥ 6 weeks) was 35.7% (15/42) in all evaluable patients (Table 4 and Supplementary Figs. 1 and 2). As shown in Table 4, at the 1–20 mg, 30 mg and 40 mg dose levels, the confirmed ORR was 10.0%, 11.1% and 15.8% respectively, the median progression-free survival (PFS) was 47 days, 79 days and 121 days, respectively, and the median duration of response (DoR) was 77 days, 80 days and 152 days respectively, which indicated a numerically higher ORR, PFS and DoR at 40 mg among these dose levels.

## Exploratory analysis

Twenty-eight out of 42 evaluable patients had *PIK3CA* mutations, among whom the confirmed ORR was 14.3% (4/28) with a confirmed

**Table 2 | Summary of treatment-related adverse events**

| n (%) | DL1–4 CYH33, 1–20 mg QD n = 10 | DL5 CYH33, 30 mg QD n = 12 | DL6 CYH33, 40 mg QD n = 24 | All patients CYH33, 1–60 mg QD n = 51 |
|---|---|---|---|---|
| TRAEs leading to dose interruption | 2 (20.0) | 10 (83.3) | 19 (79.2) | 36 (70.6) |
| TRAEs leading to dose reduction | 1 (10.0) | 0 | 5 (20.8) | 6 (11.8) |
| TRAEs leading to dose discontinuation | 0 | 1 (8.3) | 2 (8.3) | 4 (7.8) |
| TRAEs leading to death | 0 | 0 | 0 | 0 |
| Treatment-related serious adverse events | 0 | 1 (8.3) | 4 (16.7) | 6 (11.8) |
| TRAEs occurring in ≥ 10% of patients | 7 (70.0) | 12 (100) | 23 (95.8) | 47 (92.2) |
| Hyperglycemia | 7 (70.0) | 11 (91.7) | 23 (95.8) | 46 (90.2) |
| Decreased appetite | 3 (30.0) | 4 (33.3) | 12 (50.0) | 21 (41.2) |
| Nausea | 3 (30.0) | 6 (50.0) | 9 (37.5) | 19 (37.3) |
| Weight loss | 1 (10.0) | 4 (33.3) | 9 (37.5) | 16 (31.4) |
| Diarrhea | 3 (30.0) | 2 (16.7) | 7 (29.2) | 15 (29.4) |
| Peripheral edema | 1 (10.0) | 4 (33.3) | 8 (33.3) | 13 (25.5) |
| Vomiting | 3 (30.0) | 3 (25.0) | 6 (25.0) | 13 (25.5) |
| Rash | 1 (10.0) | 2 (16.7) | 6 (25.0) | 9 (17.6) |
| Fatigue | 1 (10.0) | 2 (16.7) | 6 (25.0) | 9 (17.6) |
| Facial edema | 0 | 4 (33.3) | 4 (16.7) | 8 (15.7) |
| Mouth ulceration | 1 (10.0) | 0/0 | 5 (20.8) | 6 (11.8) |
| Hyponatremia | 0 | 4 (33.3) | 2 (8.3) | 6 (11.8) |
| Grade ≥ 3 TRAEs in all patients | 2 (20.0) | 7 (58.3) | 19 (79.2) | 33 (64.7) |
| Hyperglycemia | 2 (20.0) | 7 (58.3) | 17 (70.8) | 30 (58.8) |
| Peripheral edema | 0 | 0 | 4 (16.7) | 4 (7.8) |
| Diarrhea | 0 | 0 | 1 (4.2) | 2 (3.9) |
| Decreased appetite | 0 | 0 | 1 (4.2) | 1 (2.0) |
| Mouth ulceration | 0 | 0 | 1 (4.2) | 1 (2.0) |
| Face oedema | 0 | 0 | 1 (4.2) | 1 (2.0) |
| Fatigue | 0 | 0 | 1 (4.2) | 1 (2.0) |
| Generalized oedema | 0 | 1 (8.3) | 0 | 1 (2.0) |
| Rash | 0 | 0 | 1 (4.2) | 1 (2.0) |
| Dizziness | 0 | 0 | 1 (4.2) | 1 (2.0) |
| White blood cell count decreased | 0 | 0 | 1 (4.2) | 1 (2.0) |
| Diabetic ketosis | 0 | 1 (8.3) | 0 | 1 (2.0) |
| Nausea | 0 | 0 | 0 | 1 (2.0) |
| Blood glucose increased | 0 | 0 | 0 | 1 (2.0) |
| Electrocardiogram QT prolonged | 1 (10.0) | 0 | 0 | 1 (2.0) |

Adverse events were categorized and graded using National Cancer Institute Common Terminology Criteria for Adverse Events (NCI-CTCAE) v4.03.

*DL* dose level, *QD* once-daily, *TRAE* treatment-related adverse event.

response observed in patients with breast cancer (2/5, 40%) and ovarian cancer (2/5, 40%), and a DCR of 46.4% (13/28). One patient with ovarian cancer (*PIK3CA* E545A mutation) who had received 2 lines of previous chemotherapy achieved PR after 5.3 weeks of treatment at the 40 mg dose level and a CR after 29.3 weeks per Response Evaluation Criteria in Solid Tumors (RECIST) 1.1, the DoR was 15.2 months. One patient with gastric cancer (*PIK3CA* E542K mutation) achieved a PR after 6 weeks of treatment at the 30 mg dose level, but had progressive disease (PD) at week 11.

## Case study

One of the patients in the 40 mg CYH33 treatment group, a 45-year-old female with breast cancer (Luminal B, HER2-, *PIK3CA* E545K mutation), had a rapid and remarkable response to CYH33. She had failed 2 lines of systemic treatment before being enrolled in this study. Baseline computed tomography (CT) imaging showed an oval outward bulging mass in the upper outer quadrant of her right breast (target lesion, Fig. 3a) and multiple lung metastases (all <1 cm, non-target lesions). On the Cycle 1 Day 8 (C1D8) visit of CYH33 treatment, the patient experienced visible shrinkage and necrosis of the right breast mass, and the necrotic mass disappeared on the C1D15 visit. Cancer ulcers subsequently formed on the skin of the right breast, which was followed by a significant reduction in the ulcer area, and skin healing. On C2D15, the first RECIST 1.1 tumor assessment classed the response as a PR, with notable shrinkage of the right breast mass into a subcutaneous patchy lesion (Fig. 3b), and the small metastatic lesions in the lung largely remained unchanged. The DoR was 22.0 weeks. However, the patient then experienced tumor progression on C8D1, whereby the mass in the upper outer quadrant of her right breast regrew rapidly (Fig. 3c). Archived hematoxylin-eosin stained tumor biopsy specimens acquired before baseline, at C4D5 (tumor response: PR), and at 4 weeks after the end of study (tumor response: PD) were reviewed (Fig. 3d–j). An abundant infiltration of immune cells (lymphocytes, histocytes, and macrophages) into the tumor was observed at C4D5 and coincided with the PR, but later disappeared when the cancer progressed. Interestingly, in tumor specimens at PD, both tumor regression and tumor growth were observed, suggesting intratumor heterogeneity with a subset of tumor cells resistant to CYH33 treatment.

## Discussion

CYH33 is a potent and highly selective PI3Kα inhibitor that has shown antitumor activity in in vitro and xenograft models of various cancers[17], particularly those harboring *PIK3CA* mutations. In this first-in-human phase Ia clinical trial, we demonstrated that once-daily CYH33 within the 1–60 mg dose range is safe and well tolerated in patients with solid tumors and the single-agent MTD is 40 mg once daily. Furthermore, our study provides evidence of the anti-tumor activity of PI3Kα inhibitors in a range of solid tumors, and especially in cancers harboring *PIK3CA*-specific mutations. We also provide the description of the PK profile of CYH33, with a median $t_{1/2}$ of around 20 h, which justifies the rationale for once-daily administration. Moreover, we showed that $AUC_{0-24h}$ and $C_{max}$ increased in a dose-dependent manner, indicating a linear PK profile.

The PI3K pathway has long been known to play a central role in tumor cell proliferation and survival[25]. However, it has taken decades to demonstrate the clinical benefits of PI3K inhibition in solid tumors[26]. Initially, PI3K inhibitors were predominantly developed in the form of pan-PI3K inhibitors[27,28], and their efficacy was limited by numerous side-effects and the ability to only target hematological tumors[26]. Next-generation isoform-selective PI3K inhibitors were more recently developed to overcome these limitations[29]. Alpelisib, a first-in-class oral selective PI3Kα inhibitor, has been approved for the treatment of patients with HR+/HER2−, *PIK3CA*-mutated advanced or metastatic breast cancer, when used in combination with fulvestrant[14]. The success of alpelisib supports the concept that an isoform-specific PI3K inhibitor could overcome the limitations of pan-PI3K inhibitors, bringing potential clinical benefits to patients with solid tumors[30].

Our study revealed that CYH33 is generally well tolerated. Hyperglycemia is a predictable on-target toxicity of PI3Kα inhibitors[31], and it was the most common TRAE observed during CYH33 treatment. However, CYH33-induced hyperglycemia was generally well managed with anti-hyperglycemic medication and was reversible after treatment discontinuation. Furthermore, the incidence rate for rash in our

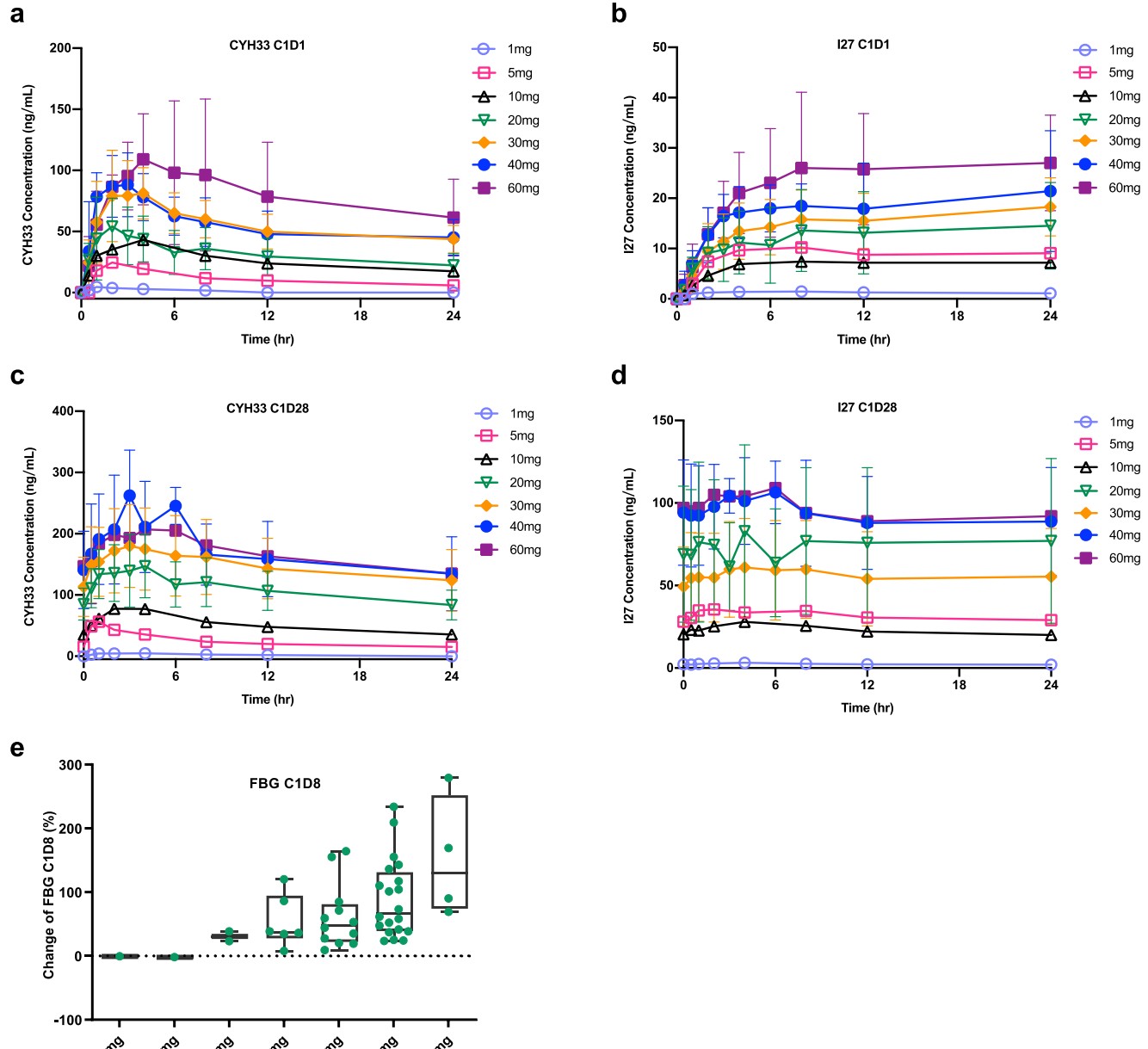

**Fig. 2 | Pharmacokinetic results. a** CYH33 and **b** its active metabolite I27 Concentration-Time Profiles (Mean ± SD) on Cycle 1 Day 1 (1 mg, *n* = 1; 5 mg, *n* = 1; 10 mg, *n* = 2; 20 mg, *n* = 5; 30 mg, *n* = 12; 40 mg, *n* = 3; 60 mg, *n* = 4); **c** CYH33 and **d** its active metabolite I27 Concentration-Time Profiles (Mean ± SD) on Cycle 1 Day 28 (1 mg, *n* = 1; 5 mg, *n* = 1; 10 mg, *n* = 2; 20 mg, *n* = 6; 30 mg, *n* = 9; 40 mg, *n* = 7; 60 mg, *n* = 2); **e** Pharmacodynamic results showing changes of FBG levels from baseline on Cycle 1 Day 8 (1 mg, *n* = 1; 5 mg, *n* = 1; 10 mg, *n* = 2; 20 mg, *n* = 6; 30 mg, *n* = 12; 40 mg, *n* = 20; 60 mg, *n* = 4). The box represents 25th, 50th, and 75th percentiles of observed values; the whiskers represent the minimum and maximum values; green dots represent individual values. hr hour, FBG fasting blood glucose, SD standard deviation.

study was 36.7% at CYH33 40 mg QD, which was lower than 64.0% reported for alpelisib[32,33]. Rugo et al.[32] have previously shown that the incidence of alpelisib-induced rash could be rapidly lowered from 64.0% to 24.6% with prophylactic anti-rash medication within the first 4–8 weeks of alpelisib treatment. Thus, the use of prophylactic anti-rash medication for rash prevention could be considered in future studies of CYH33. In addition, the frequency of diarrhea at CYH33 40 mg QD was also found to be lower than reported for alpelisib (29.3% vs. 57.7%)[32,33]. Considering the safety and efficacy profile of CYH33 at 40 mg QD, 40 mg QD is selected as the RP2D of CYH33.

A preliminary evaluation of the anti-tumor efficacy of CYH33 suggested that this PI3Kα inhibitor is effective for treating several types of *PIK3CA*-mutant solid tumors such as breast, ovarian,

and gastric cancers. This finding is consistent with the fact that *PIK3CA* mutation plays an oncogenic role in these types of solid tumors[9]. In the present study, the longest DoR was 15.2 months, in an ovarian cancer patient. However, treatment with PI3K inhibitors has been previously shown to trigger acquired drug resistance[26]. Thus, the duration of the anti-tumor response is likely to be short for the majority of tumor types. Whether treatment with a single PI3Kα inhibitor can bring significant clinical benefit to patients with solid tumors needs to be further evaluated. Furthermore, combinatorial treatment strategies may improve the efficacy of PI3Kα inhibitors. We observed an interesting phenomenon in one patient with breast cancer who achieved a PR in whom CYH33 treatment induced lymphocyte infiltration in tumor tissue. This phenomenon is in accordance with a prior research finding

**Table 3 | Summary of CYH33 PK parameters on Cycle 1 Day 1ᵃ**

| Dose | T_max, h, median (min-max) | C_max, ng/mL | AUC_0-24h, h*ng/mL | AUC_0-∞, h*ng/mL | t_1/2, h | V/F, L | CL/F, L/h |
|---|---|---|---|---|---|---|---|
| 1 mg (n = 1) | 1 | 4.6 | 26.0 | 36.8 | NA | NA | NA |
| 5 mg (n = 1) | 2 | 24.6 | 265.7 | 429.2 | 19.2 | 322.8 | 11.6 |
| 10 mg (n = 2) | 4 (4–4) | 43.1 (0) | 623.5 (5) | 1092.2 (12) | 20.2 (35) | 263.2 (24) | 9.2 (11) |
| 20 mg (n = 5) | 2 (1–2) | 56.3 (46) | 763.0 (47) | 2186.5 | 20.8 | 274.2 | 9.1 |
| 30 mg (n = 12) | 3.5 (2–8) | 96.5 (30) | 1303.3 (23) | NA | NA | NA | NA |
| 40 mg (n = 3) | 2 (2–3) | 97.9 (28) | 1317.9 (31) | NA | NA | NA | NA |
| 60 mg (n = 4) | 4 (1–8) | 123.1 (32) | 1871.8 (49) | NA | NA | NA | NA |

All data are summarized as mean, CV% (coefficient of variation) unless indicated.

%AUC_ex percentage of AUC0- ∞ extrapolated to infinity, AUC_0-24 area under the plasma concentration-time curve from time 0 to 24 h, AUC_0-∞ area under the plasma concentration-time curve from time 0 to infinity, C_max maximum plasma drug concentration, CL/F apparent total clearance of the drug from plasma after oral administration, PK pharmacokinetic, t_1/2 terminal elimination half-life, T_max maximum plasma drug concentration, V/F apparent volume of distribution.

ᵃAt all dose levels, AUC0-∞, t1/2, V/F, CL/F were only calculated when %AUCex is <30%.

**Table 4 | Summary of clinical efficacy in evaluable analysis set**

| Clinical efficacyᵃ | DL1–4 CYH33, 1–20 mg QD n = 10 | DL5 CYH33, 30 mg QD n = 9 | DL6 CYH33, 40 mg QD n = 19 | All patients CYH33, 1–60 mg QD n = 42 |
|---|---|---|---|---|
| CR, n (%) | 0 | 0 | 1 (5.3) | 1 (2.4) |
| PR, n (%) | 1 (10.0) | 1 (11.1) | 2 (10.5) | 4 (9.5) |
| SD, n (%) | 2 (20.0) | 6 (66.7) | 9 (47.4) | 18 (42.9) |
| Confirmed ORRᵇ, n (%)[95% CI] | 1 (10.0) | 1 (11.1) | 3 (15.8) | 5 (11.9) |
| | [0.25, 44.5] | [0.28, 48.25] | [3.38, 39.58] | [3.98, 25.63] |
| DCR (CR + PR + SDᶜ), n (%) | 3 (30.0) | 1 (11.1) | 10 (52.6) | 15 (35.7) |
| CBR, n (%) | 2 (20.0) | 1 (11.1) | 3 (15.8) | 6 (14.3) |
| mPFSᵈ, days [95% CI] | 47 | 79 | 121 | 79 |
| | [36.00,86.00] | [25.00, 97.00] | [42.00, 197.00] | [42.00, 116.00] |
| mDoRᵈ, days | 77 | 80 | 152 | 80 |
| [95% CI] | [NE, NE] | [NE, NE] | [64.00, NE] | [64.00, NE] |

DL dose level, QD once daily, CR complete response, PR partial response, SD stable disease, ORR objective response rate, DCR disease control rate, CBR clinical benefit rate, mPFS median progression-free survival, mDoR median duration of response, CI confidence interval, NE not evaluated.

ᵃResponse assessed using RECIST 1.1.

ᵇDefined as confirmed CR + PR, excluding two patients without confirmed responses.

ᶜDefined as SD lasting ≥ 6 weeks, excluding eight patients whose SD lasted <6 weeks.

ᵈPFS and DoR were evaluated using the Kaplan–Meier estimates.

that PI3Kα inhibition with CYH33 triggers antitumor immunity in breast cancer by activating CD8 + T cells[18], suggesting a potential synergistic effect of combining CYH33 with immune checkpoint inhibitors. In addition, several studies have suggested that PI3Kα inhibitors may exhibit synthetic lethality when used in combination with other anti-tumor therapies such as poly ADP-ribose polymerase (PARP) inhibitors[34], BET inhibitors[35], or fulvestrant[15,32]. Thus, studies focusing on the anti-cancer properties of CYH33, either as a single agent or in combination with other anti-tumor drugs, are currently ongoing to further evaluate its safety and efficacy in larger cohorts of cancer patients. This includes a global phase Ib study of CYH33 in combination with olaparib in advanced solid tumors (NCT04586335) and a phase Ib study of CYH33 in combination with endocrine therapy with or without palbociclib in patients with HR+, HER2− advanced breast cancer (NCT04856371).

The limitations of this study include the single-center design of the phase Ia stage of the study which may reduce the generalizability of the findings, but was implemented to enable optimal safety oversight of the early stages of the trial. In addition, this study only included Chinese patients, and the results are therefore not generalizable to other populations, with further investigation required.

In conclusion, the present study provides the first-in-human clinical evidence for CYH33 and shows that this drug is generally well tolerated with promising anti-tumor efficacy in several types of advanced solid tumors. The single-agent MTD for CYH33 was established as a once-daily dose of 40 mg. Based on these findings, further large-scale investigations of CYH33 and other isoform-selective PI3Kα inhibitors are warranted.

## Methods

The study protocol (see Supplementary Note 1) was approved by the ethics committee and institutional review board of Sun Yat-sen University Cancer Center and complied with the Declaration of Helsinki and guidelines for Good Clinical Practice, as defined by the International Conference on Harmonization. All patients provided written informed consent.

### Study design

This is a phase I clinical study (ClinicalTrials.gov identifier: NCT03544905), including a phase Ia study of oral CYH33 monotherapy in patients with advanced solid tumors, and a phase Ib study of oral CYH33 monotherapy focused on several specific tumors. Here we report the results of the phase Ia study conducted at Sun Yat-sen University Cancer Center. The phase Ia study included dose-escalation stage and dose-expansion stage. The primary objectives in phase Ia were to determine the safety, tolerability and MTD of oral CTH33 monotherapy in patients with advanced solid tumors who have failed or cannot tolerate standard treatment or currently have no standard

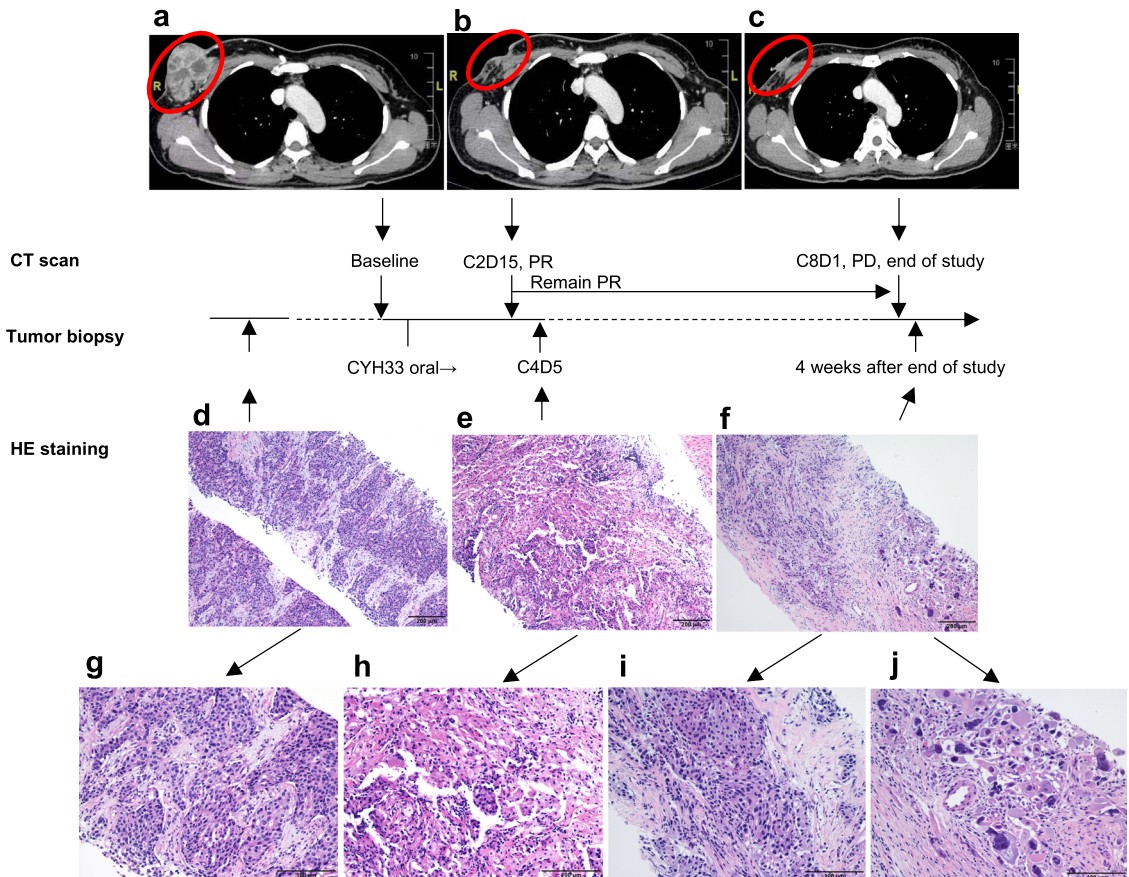

**Fig. 3 | Representative CT images and HE stained biopsy sections from a patient in the 40 mg CYH33 treatment group with advanced breast cancer (Luminal B, HER2-, *PIK3CA* E545K mutation) who achieved a rapid and robust treatment response. a** Baseline CT image (September 18, 2019): a mass (red circle) was detected in the upper outer quadrant of the right breast on enhanced CT. **b** Cycle 2 Day 15 CT image (November 25, 2019): the mass (red circle) had reduced on enhanced CT compared with baseline. **c** End of study visit CT image (April 24, 2020): the mass (red circle) had increased on enhanced CT. HE stained biopsy sections from pre-baseline (**d** scale bar = 200 µm; **g** scale bar = 100 µm), at partial response (**e** scale bar = 200 µm; **h** scale bar = 100 µm) and at disease progression (**f** scale bar = 200 µm; **i** and **j** scale bar = 100 µm). PR partial response, PD progressive disease, HER2- human epidermal growth factor receptor 2 negative, CT computed tomography, HE hematoxylin-eosin.

treatment, and determine the RP2D; while the secondary objectives were to assess the preliminary efficacy of oral CYH33 monotherapy, to determine the PK characteristics of CYH33 and its metabolite (I27) after single and continuous oral administration of CYH33. The secondary outcomes also included evaluation of changes in PD biomarkers in response to CYH33 treatment (including proteomics of blood specimens at baseline and during treatment, immunocyte subpopulation ratios and molecular markers, cell free DNA in peripheral blood, and blood glucose). However, in compliance with the local human genetic resource regulations and procedures, peripheral blood samples were not obtained for PD biomarker analysis, therefore, only fasting blood glucose was obtained and used as a PD biomarker for CYH33 treatment. The exploratory objective was the relationship between the efficacy of CYH33 treatment and tumor biomarker status.

**Dose-escalation stage.** An accelerated titration design (ATD) and an mTPI-2 design were adopted to guide toxicity monitoring and dose escalation. The target DLT rate was 30%, with the acceptable toxicity probability interval of (0.25, 0.35). For the first two preset doses of 1 mg and 5 mg CYH33 in the ATD phase, if there were no grade ≥2 TRAEs during the DLT observation period, 100% dose escalation was allowed in the subsequent dose group. After entering the mTPI-2 phase, the SMC guidance was followed to determine the dose level, dosing frequency, and number of subjects in the next dose group, based on the previously obtained safety and PK data for that dose

group. If necessary, it was permitted to add an additional dose level between two explored dose levels for further investigation.

**Dose-expansion stage.** During the phase Ia study, the safe and probably effective dose will be delivered to the expansion cohort to further investigate the safety, tolerability and efficacy of this dose level, while the dose escalation study can be continued simultaneously. If necessary, it is permitted to insert a new dose level between two explored dose levels for exploration. The subjects to be enrolled in the expansion cohort are patients with advanced solid tumors with *PIK3CA* gene mutations. When there are two or more dose groups at dose expansion simultaneously, the allocation ratio of the number of subjects in each dose group should be as close as possible to the ratio of ORR in each dose group. In principle, when the cumulative ORR at a certain dose level is <10%, the number of subjects in this dose group will be determined by the SMC. A total of approximately 60 patients are expected to be enrolled in the whole phase Ia study.

The dose escalation stage of the CYH33-101 phase Ia study was designed to include ascending doses from 1 to 240 mg QD, and to enroll approximately 30 patients based on an ATD and an mTPI-2 design. The target number of enrollment for the subsequent dose expansion stage was around 30 patients. The final number was determined by the SMC based on the safety, tolerability, PK and/or efficacy data in each dose escalation and dose expansion cohort. In the dose escalation stage, the MTD was determined to be 40 mg QD, which

was lower than estimated, thus only 19 patients were enrolled (1 mg: $n = 1$; 5 mg: $n = 1$; 10 mg: $n = 2$; 20 mg: $n = 2$; 40 mg: $n = 8$; 60 mg: $n = 5$). After the MTD was defined, additional 32 patients were enrolled in the dose expansion stage to further evaluate the safety, tolerability, PK and biologic activity of CYH33. The primary outcomes of phase Ia study had been achived with enrolled 51 patients, therefore the SMC decided to discontinue the enrollment of phase Ia.

## Patients

This study enrolled Chinese patients (aged > 18 years) with histopathologically- or cytologically-confirmed, locally advanced or metastatic solid tumors, who had failed, or could not tolerate standard treatment or had no standard treatment (including patients who rejected chemotherapy). Other key eligibility criteria included measurable disease, defined according to RECIST 1.1, ECOG performance status ≤ 1, and adequate organ function at screening (including fasting plasma glucose < 126 mg/dL [7.0 mmol/L]). Key exclusion criteria were previously failed treatment with PI3K, AKT, or mTOR inhibitors, as well as the presence of central nervous system metastases or clinically significant organ dysfunction. In the dose-escalation stage, patients with solid tumors were enrolled, regardless of their *PIK3CA* mutation status. In the dose-expansion stage, patients with *PIK3CA* mutations were enrolled to evaluate the sensitivity of *PIK3CA*-mutated tumors to CYH33 treatment. The exact dates of first and last patient enrollment were July 13, 2018 and March 29, 2021, respectively.

## Study treatment

Eligible patients received CYH33 tablets orally once daily, either in a fasted state or 2 h after a meal, with a tentative 28-day treatment cycle.

## Study endpoints

The primary endpoints of the phase Ia included (i) the type and frequency of treatment emergent adverse events (TEAEs) which were assessed per National Cancer Institute Common Terminology Criteria for Adverse Events (NCI-CTCAE) v4.03, laboratory results, electrocardiogram (ECG), cardiac imaging, physical examination findings (including vital signs, weight, and ECOG scores), (ii) number and proportion of subjects who occurred DLTs in each dose group during dose-escalation stage, (iii) the MTD, if the MTD was not observed, the RP2D will be determined through PK/PD data, safety and preliminary efficacy. The secondary endpoints included preliminary efficacy assessments based on ORR, PFS, DoR, DCR, and clinical benefit rate, assessed per RECIST 1.1, PK parameters for CYH33 and its metabolite I27, and changes in PD biomarkers in response to CYH33 treatment.

A DLT was defined as a TRAE or laboratory abnormality that occurred within 28 days after the first dose of CYH33 in the dose-escalation stage (and within 35 days of the first dose for patients receiving a single administration; i.e., 7 days of the single dose phase + 28 days of Cycle 1 of the continuous dose phase) and met any of the following criteria as per NCI-CTCAE v4.03: (1) grade 2 hyperglycemia, in which the blood glucose level failed to return to normal FBG or baseline levels within 14 days after appropriate anti-diabetic treatment, grade 3 or asymptomatic grade 4 hyperglycemia, which did not return to grade ≤ 2 hyperglycemia within 7 days after suspension of CYH33 and adequate/appropriate anti-diabetic treatment, or symptomatic grade 4 hyperglycemia; in all cases, the FBG level was retested within 24 h for confirmation; and (2) grade 3 thrombocytopenia with a significant bleeding tendency, persisting for > 7 days and failing to return to grade ≤ 2 or baseline level within 7 days after the study drug was suspended, grade 3 neutropenia with fever persisting for > 7 days, grade 4 neutropenia, or other grade 4 hematological toxicities; and (3) grade ≥ 3 non-hematological (except hyperglycemia) toxicities. Patients who had received at least 75% of the assigned doses within 28 days of the first cycle of the continuous dose phase were included for DLT analysis.

## PK and PD assays

Blood samples for PK evaluation were collected at: (1) during the single dose phase: on day 1 pre-dose and 0.5, 1, 2, 4, 8, 12, 24, 36, 48, and 72 h post-dose; (2) during the continuous dose phase: C1D1/Cycle 1 Day 28 pre-dose, and 0.5, 1, 2, 3, 4, 6, 8, 12, and 24 h post-dose; and C1D8/Day 15 pre-dose. Plasma concentrations of CYH33 and I27 were analyzed using a liquid chromatography-tandem mass spectrometry (LC/MS/MS) assay (Shimadzu Corporation, LC-30AD) with a lower limit of quantification (LLOQ) of 2.585 ng/mL and 1.0 ng/mL for CYH33 and I27, respectively. The following PK parameters were calculated for CYH33 and I27 concentration-time data using the Phoenix Winnonlin software v8.3: $C_{max}$, $T_{max}$, area under the plasma concentration-time curve from time 0 to infinity, and 24 h ($AUC_{0\text{-}last}$, $AUC_{0\text{-}\infty}$, and $AUC_{0\text{-}24\,h}$), $t_{1/2}$, total body clearance (CL/F), and volume of distribution (V/F). Blood samples for measuring the FBG level were collected within 30 min before dosing on C1D1, C1D15, and on the first day of subsequent treatment cycles.

## Statistics and reproducibility

Sample size considerations. According to the mTPI-2 study design[22], approximately 3–5 patients were required at each dose level during dose escalation, thus 30 patients were planned in the dose escalation stage. A approximately 10 patients were planned to be enrolled in each expansion cohort, thus 30 patients were estimated to be enrolled in the dose expansion stage.

The analysis of the phase Ia study was pre-planned and reviewed by the SMC. The Kaplan–Meier method was used to summarize time-to-event endpoints. Response rates were summarized using a point estimate and 95% Clopper–Pearson confidence interval. The full analysis set (FAS) included all study patients who received at least one dose of study drug and was used for the analysis of baseline characteristics. The safety analysis set included all study patients who had received at least one dose of the study drug and was used for all safety analysis. The efficacy evaluable analysis set included patients who received at least one dose of CYH33 and had baseline tumor assessment data and at least one post-baseline tumor assessment. All statistical analyses were performed using SAS software v9.3 (Cary, North Carolina, USA). No data were excluded from the analyses. As a phase Ia, open label, dose-escalation and -expansion study, the study was not randomized, and the investigators were not blinded to allocation during trial and outcome assessment.

## Reporting summary

Further information on research design is available in the Nature Portfolio Reporting Summary linked to this article.

## Data availability

The study protocol is available as Supplementary Note 1 and the statistical analysis plan as Supplementary Note 2 in the Supplementary Information file. Clinical data are not publicly available due to involving patient privacy, but can be accessed on request from the corresponding author R.-H.X. for 10 years; individual de-identified participant data will be shared. The remaining data are available within the Article, Supplementary Information or Source Data file. Source data are provided with this paper.

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

## Acknowledgements

We would like to thank all patients and their families as well as the study investigators and personnel for participating in this trial. Jake Burrell PhD (Rude Health Consulting) provided editorial support for this manuscript. This study was supported by grants from the Science and Technology Commission of Shanghai Municipality (STCSM No.15DZ193050 [Du] and No.19431905100 [Du]) and sponsored by Haihe Biopharma Co. Ltd, China.

## Author contributions

R.-H.X. is the leading PI, who designed and conducted this trial and contributed to the writing and review of the manuscript. X.-L.W., F.-R.L., J.-H.L., H.-Y.Z., Y.Z., Z.-Q.W., M.-Z.Q., F.X., Y.-X.S., D.-S.W., and F.-H.W. are the investigators of this trial, who contributed to manuscript writing and review. Q.-Q.Y. and Y.-W.D. are the clinical research physicians, who contributed to medical monitoring and data analysis as well as review of the manuscript content. All authors reviewed the final version of this manuscript and approved it for submission.

## Competing interests

The authors declare the following competing interests: Q.-Q.Y. and Y.-W.D. are employed by Haihe Biopharma Co., Ltd. The rest of authors have no conflicts of interest to declare.
