## [Peer Review File · Nature Communications]

Reviewers' Comments:

Reviewer #1:

Remarks to the Author:

This is a phase Ia dose escalation and expansion study investigating CYH33, a novel oral PI3K67 α inhibitor in patients with solid tumors. The dose escalation part was based on the modified TPI-2 (mtpi-2) design, whereas the design consideration for the dose expansion part is unclear.

- 1) Number and proportion of patients harboring PIK3CA should be reported.
- 2) The numbers of patients harboring PIK3CA mutations do not match throughout the paper. For example, on page 5 lines 84-88, it was stated that 32 patients harboring PIK3CA mutations were included in the dose-expansion stage. However, in the abstract and on page 9 lines 156-157, only 28 patients were harboring PIK3CA mutations.
- 3) According to the study protocol, only patients with PIK3CA mutations should be enrolled in the dose expansion phase. However, it seems that patients with unknown mutation status were actually enrolled as well. If this is the case, please explain why the protocol was violated.
- 4) In Table 1, please report the PIK3CA status.
- 5) On page 14 dose-escalation stage, please indicate the target DLT rate used in the mTPI-2 design.
- 6) On page 15, the description of the stage design in the dose-expansion stage is rather vague. It seems that multiple dose levels had been explored in the dose expansion phase. However, it's unclear about the statistical method the selection and assignment were based.
- 7) In the swimmer plot please indicate the PIK3CA status.

Reviewer #2:

Remarks to the Author:

The fact that the 42 evaluable patients used to determine response rate was out of 51 total patients should be included in the abstract.

Page 6, Under Dose-limiting toxicity it states that the 30 mg dose level was skipped. Yet, in Table 4 it lists 9 patients who have received the 30 mg dose. One of these two needs to be corrected.

Under efficacy, page 9; The sentences beginning on line 167 with "in 28 evaluable..." are somewhat confusing because while it states there were 4/28 responses in patients with PIK3CA mutations, it then lists the 5 overall responses (2 ovarian, 2 breast, 1 colorectal). Presumably one of these was a patient with undefined PIK3CA status. But, in any case, this needs to be clarified so that it is clear what the response rates were in each disease by PIK3CA status.

Page 10: Line 195--Was there any further analysis to define the lymphocyte subsets seen on the biopsies to evaluate changes in these over time?

Under Discussion: The sentence beginning on line 236, page 12--the wording should be changed from "a range" to several types of PIK3CA-mutant solid tumors since the responses were seen in just 3 types (breast, ovarian, and colorectal). In addition, a comment about the response rate appearing to be lower in colorectal cancer should be included but with the caveat of small patient numbers.

Page 10, under Study endpoints, also define the secondary endpoints of the study.

Page 25, in the image for Fig 2, it is not possible to see the metastatic lesion (at least in the printed document. This should be checked again to make sure the lesion is visible to the reader.

Reviewer #3:

Remarks to the Author:

This manuscript by Wei and colleagues describes the first in man study of a novel PI3K α inhibitor. This study shows acceptable pharmacokinetics for this compound and advises on a recommended Phase II dose based on tolerability.

The study confirms the benefit targeting the α isoform in patients with mutations and confirms activity in patients with PIK3CA mutations. Of note is the fact that this study was carried out in a Chinese populations.

The study is well carried out and it is exciting to see another PI3K α selective inhibitor in clinical study but there is nothing new here (apart from a novel compound) to warrant publication in

nature communication.

Of note is the fact that the biomarker used is fasting glucose which is a distal biomarker on the pathway. From this, it is difficult to evaluate the real target engagement.

Multiple studies with PI3K inhibitors have used P-AKT in tumours and peripheral blood mononuclear cells and this should have been done here as the PKPD relationships are easily established and known.

Overall, this is an interesting study but this work should be submitted to a different more appropriate journal.

Response Letter

Reviewer #1 - Biostatistics, clinical trials - (Remarks to the Author):

This is a phase Ia dose escalation and expansion study investigating CYH33, a novel oral PI3K α inhibitor in patients with solid tumors. The dose escalation part was based on the modified TPI-2 (mTPI-2) design, whereas the design consideration for the dose expansion part is unclear.

1) Number and proportion of patients harboring *PIK3CA* should be reported.

Reply: Thanks for your valuable comments. we revised the manuscript accordingly.

Change in the text:

Page 3 lines 31-38: “A total of 51 patients including 36 (70.6%) patients with *PIK3CA* mutations Among 36 patients harboring *PIK3CA* mutations (4 patients in the dose-escalation stage and 32 patients in the dose-expansion stage), 28 patients were evaluable for response,.....”

Page 5 lines 85-90: “Between July 2018 and May 2021, 51 patients (median age, 54 years) were enrolled in phase Ia study and received CYH33 (Table 1). Of these patients, 19 patients were recruited to the dose-escalation stage, with 15 (78.9%) patients having unknown *PIK3CA* mutation status and 4 (21.1%) patients harboring *PIK3CA* mutations. Thirty-two (100%) patients harboring *PIK3CA* mutations determined via local laboratory testing were included in the dose-expansion stage, 4 patients of them in the 20 mg group, 12 in the 30 mg group, and 16 in the 40 mg dose group.”

Hope above explanation and revision can answer your questions. Thanks again for your valuable comments.

2) The numbers of patients harboring *PIK3CA* mutations do not match throughout the paper. For example, on page 5 lines 84-88, it was stated that 32 patients harboring *PIK3CA* mutations were included in the dose-expansion stage. However, in the abstract and on page 9 lines 156-157, only 28 patients were harboring *PIK3CA* mutations.

Reply: Thanks for your valuable comments. Sorry for the confusing wording in the manuscript. A total of 36 patients harboring *PIK3CA* mutations were enrolled in the phase Ia study, 4 patients of them were included in the dose-escalation stage, the other 32 patients were included in the dose-expansion stage. Among 36 patients with *PIK3CA* mutations (32 patients were enrolled in the dose-expansion stage), only 28 patients conducted tumor assessment and were evaluable for response after study drug administration.

In order to avoid this confusion, we revised the manuscript as below:

Change in the text:

Page 3 lines 31-39: “A total of 51 patients including 36 (70.6%) patients with *PIK3CA* mutations Forty-two out of 51 patients were evaluable for response, the confirmed objective response rate was 11.9% (5/42). Among 36 patients harboring *PIK3CA* mutations (4 patients in the dose-escalation stage and 32 patients in the dose-expansion stage), 28 patients were evaluable for response, the confirmed objective response rate was 14.3% (4/28).”

Page 5 lines 85-90: “Between July 2018 and May 2021, 51 patients (median age, 54 years) were enrolled in phase Ia study and received CYH33 (Table 1). Of these patients, 19 patients were recruited to the dose-escalation stage, with 15 (78.9%) patients having unknown *PIK3CA* mutation status and 4 (21.1%) patients harboring *PIK3CA* mutations. Thirty-two (100%) patients harboring *PIK3CA* mutations determined via local laboratory testing were included in the dose-expansion stage, 4 patients of them in the 20 mg group, 12 in the 30 mg group, and 16 in the 40 mg dose group.”

Hope above explanation and revision can answer your questions. Thanks again for your valuable comments.

- 3) **According to the study protocol, only patients with *PIK3CA* mutations should be enrolled in the dose expansion phase. However, it seems that patients with unknown mutation status were actually enrolled as well. If this is the case, please explain why the protocol was violated.**

Reply: Thanks for your valuable comments. Sorry for the confusion. In fact, no protocol violation occurred regarding the status of *PIK3CA* mutations. In the dose expansion stage, all 32 enrolled patients had *PIK3CA* mutations.

In order to avoid this confusion, we revised the manuscript as below:

Change in the text:

Page 5 lines 85-90: “Between July 2018 and May 2021, 51 patients (median age, 54 years) were enrolled in phase Ia study and received CYH33 (Table 1). Of these patients, 19 patients were recruited to the dose-escalation stage, with 15 (78.9%) patients having unknown *PIK3CA* mutation status and 4 (21.1%) patients harboring *PIK3CA* mutations. Thirty-two (100%) patients harboring *PIK3CA* mutations determined via local laboratory testing were included in the dose-expansion stage, 4 patients of them in the 20 mg group, 12 in the 30 mg group, and 16 in the 40 mg dose group.”

Hope above explanation and revision can answer your questions. Thanks again for your valuable comments.

- 4) **In Table 1, please report the *PIK3CA* status.**

Reply: Thanks for your good suggestions. We revised the Table 1 accordingly.

Change in the text:

Table 1. Patient demographics at baseline

Characteristic	DL1-4 CYH33, 1-20 mg QD n = 10	DL5 CYH33, 30 mg QD n = 12	DL6 CYH33, 40 mg QD n = 24	All patients CYH33, 1-60 mg QD N = 51
Age (years) median (range)	55.5 (47, 67)	54.5 (34, 70)	49.0 (26, 73)	54.0 (26, 73)
Sex (n) (male/female)	6/4	4/8	9/15	22/29
ECOG status, n (%)				
0	4 (40.0)	3 (25.0)	10 (41.7)	18 (35.3)
1	6 (60.0)	9 (75.0)	14 (58.3)	33 (64.7)
Number of prior systemic regimens, n (%)				
1	2 (20.0)	7 (58.3)	10 (41.7)	19 (37.3)
2	2 (20.0)	2 (16.7)	6 (25.0)	12 (23.5)
3	2 (20.0)	0	3 (12.5)	8 (15.7)
≥ 4	4 (40.0)	3 (25.0)	5 (20.8)	12 (23.5)
Number of metastatic sites, n (%)				
1	0	1 (8.3)	7 (29.2)	9 (17.6)
2	4 (40.0)	4 (33.3)	7 (29.2)	16 (31.4)
≥ 3	6 (60.0)	7 (58.3)	10 (41.7)	26 (51.0)
Status of PIK3CA , n (%)				
Mutant	4 (40.0)	12 (100.0)	17 (70.8)	36* (70.6)
Wild type	0	0	0	0
Unknown	6 (60.0)	0	7 (29.2)	15 (29.4)

DL, dose level; ECOG, Eastern Co-operative Oncology Group; QD, once-daily.

*Three patients in the 60 mg dose group harbored *PIK3CA* mutations.

- 5) **On page 14 dose-escalation stage, please indicate the target DLT rate used in the mTPI-2 design.**

Reply: Thanks for your valuable comments. we added the statement in the manuscript as below:

Change in the text:

Page 14 line 285-286: “The target DLT rate was 30%, with the acceptable toxicity probability interval of (0.25,0.35).”

- 6) **On page 15, the description of the stage design in the dose-expansion stage is rather vague.**

It seems that multiple dose levels had been explored in the dose expansion phase. However, it 's unclear about the statistical method the selection and assignment were based.

Reply: Thanks for your valuable comments. we revised and polished the descriptions of dose expansion stage according to the study protocol, which provided more detail information about the statistical method the selection and assignment were based. Rules for the number of enrolled subjects in different dose groups were defined in the protocol “when there are no less than two dose groups at the same time of expansion, the allocation ratio of the number of subjects in each dose group should be as close to the ratio of ORR in each dose group. In principle, when the cumulative ORR at a certain dose level is <10%, the number of subjects in this dose group will be determined by SMC. A total of approximately 60 patients are expected to be enrolled in whole phase Ia study.” According to above pre-defined rules in the protocol, more patients were assigned to 40mg and 30mg dose level since higher response rate were observed in these two dose levels (detail ORR and patient number in each dose level was shown in table 4).

In order to make it more clear and easier to understand, we provide the figure of patient assignment in phase Ia study as below.

CYH33-101 Phase Ia study patient assignment

Change in text:

Page 15 line 293-302: “During Phase Ia study, the safe and probably effective dose will be delivered to the expansion cohort to further investigate the safety, tolerability and efficacy of this dose level, while the dose escalation study can be continued simultaneously. If necessary, it is allowed to insert a new dose level between two explored dose levels for exploration. The subjects to be enrolled in the expansion cohort are patients with advanced solid tumors with *PIK3CA* gene mutations. When there are two or more dose groups at dose expansion simultaneously, the allocation ratio of the number of subjects in each dose group should be as close to the ratio of ORR in each dose group. In principle, when the cumulative ORR at a certain dose level is <10%, the number of subjects in this dose group will be determined by SMC. A total of approximately 60 patients are expected to be enrolled in whole phase Ia study.”

Hope above explanation and revision can answer your questions. Thanks again for your valuable comments.

7) In the swimmer plot please indicate the *PIK3CA* status.

Reply: Thanks for your good suggestions. We revised the swimmer plot accordingly.

Change in the text:

Reviewer #2 - Biostatistics, clinical trials - (Remarks to the Author):

- 1) **The fact that the 42 evaluable patients used to determine response rate was out of 51 total patients should be included in the abstract.**

Reply: Thanks for your valuable comments. we revised the abstract accordingly.

Change in the text: Page 3 line 35: “Forty-two out of 51 patients were evaluable for response,”

- 2) **Page 6, Under Dose-limiting toxicity it states that the 30 mg dose level was skipped. Yet, in Table 4 it lists 9 patients who have received the 30 mg dose. One of these two needs to be corrected.**

Reply: Thanks for your valuable comments. Sorry for the confusion. In fact, we escalated the dose from 20 mg to 40 mg first in the dose escalation stage since no DLT was observed at 20 mg dose level, and the 40 mg didn't exceed MTD (DLT rate was 16.7%, 1/6). As the study protocol clarified that “If necessary, it is allowed to insert a new dose level between two explored dose levels for exploration.”, thus the safety monitoring committee (SMC) decided to insert one dose level (30 mg) between 20 mg and 40 mg to further explore in the dose expansion stage. Refer to the figure of patient assignment in phase Ia study as below.

Hope above explanation can answer your questions. Thanks again for your valuable comments.

CYH33-101 Phase Ia study patient assignment

- 3) **Under efficacy, page 9; The sentences beginning on line 167 with "in 28 evaluable..." are**

somewhat confusing because while it states there were 4/28 responses in patients with *PIK3CA* mutations, in then lists the 5 overall responses (2 ovarian, 2 breast, 1 colorectal). Presumably one of these was a patient with undefined *PIK3CA* status. But, in any case, this needs to be clarified so that it is clear what the response rates were in each disease by *PIK3CA* status.

Reply: Thanks for your valuable comments. One colorectal cancer patient with unknown *PIK3CA* mutation status achieved PR in dose escalation, the other 4 patients with *PIK3CA* mutations were from dose expansion. In order to avoid this confusion, we revised the efficacy part in the manuscript as below:

Change in the text:

Page 9 and Page 10 line 158-180: “Of the 51 patients in the study, 42 patients were included in the final evaluable analysis set (28 harboring *PIK3CA* mutations and 14 with unknown *PIK3CA* mutation status); 9 patients missed their post-baseline assessments due to AEs or voluntary withdrawal. As of the data cut-off date 16th July 2021, half of the 42 evaluable patients experienced a shrinkage of the target lesions compared with baseline. Five patients achieved confirmed tumor response including 1 complete response (CR) and 4 partial responses (PR), of which 4 patients had *PIK3CA* mutations in the dose expansion stage and one colorectal cancer patient had unknown *PIK3CA* mutation status in the dose escalation stage. The confirmed objective response rate (ORR; CR + PR) was 11.9% (5/42; 95% confidence interval [CI], 3.98–25.63) and the disease control rate (DCR; CR + PR + SD \geq 6 weeks) was 35.7% (15/42) in all evaluable patients (Table 4 and Supplementary Figure 1 and 2). As shown in Table 4, at 1-20 mg, 30 mg and 40 mg dose levels, the confirmed ORR was 10%, 11.1% and 15.8% respectively, the median PFS was 47 days, 79 days and 121 days respectively, and the median DoR was 77 days, 80 days and 152 days respectively, which indicated a numerically higher ORR, PFS and DoR at 40 mg among these dose levels.

Twenty-eight out of 42 evaluable patients had *PIK3CA* mutations, the confirmed ORR was 14.3% (4/28) with confirmed response observed in breast cancer (2/5, 40%) and ovarian cancer (2/5, 40%), and the DCR was 46.4% (13/28). One patient with ovarian cancer (*PIK3CA* E545A mutation) who had received 2 lines of previous chemotherapy achieved PR after 5.3 weeks of treatment at 40 mg dose level and CR after 29.3 weeks per Response Evaluation Criteria in Solid Tumors (RECIST) 1.1, and the duration of response (DoR) lasted 15.2 months. One patient with gastric cancer (*PIK3CA* E542K mutation) achieved PR after 6 weeks of treatment at 30 mg dose level, but had progressive disease at week 11.”

- 4) Page 10: Line 195--Was there any further analysis to define the lymphocyte subsets seen on the biopsies to evaluate changes in these over time?**

Reply: Thank you for your valuable comments. we didn't have enough biopsy tumor samples to perform lymphocyte subsets staining, thus no further analysis was available.

We really appreciate your advice and will consider it in our further research.

- 5) **Under Discussion: The sentence beginning on line 236, page 12--the wording should be changed from "a range" to several types of *PIK3CA*-mutant solid tumors since the responses were seen in just 3 types (breast, ovarian, and colorectal). In addition, a comment about the response rate appearing to be lower in colorectal cancer should be included but with the caveat of small patient numbers.**

Reply: Thanks for your valuable comments. We revised the manuscript accordingly.

In addition, we totally agree with your comments that the patient numbers were too small in colorectal cancer and cervical cancer, it was too early to draw the conclusion that the response rate was lower in colorectal cancer and cervical cancer. Thus, we deleted the description “..... lower ORRs were observed in patients with colorectal cancer (10%, 1/10) or cervical cancer (0.0%, 0/6).” in the manuscript.

Change in the text:

Page 12 lines 240-241 “A preliminary evaluation of the anti-tumor efficacy of CYH33 suggested that this PI3K α inhibitor is effective for treating several types of *PIK3CA*-mutant solid tumors.....”

Page 14 lines 269-270 “.....shows that this drug is generally well tolerated with promising anti-tumor efficacy in several types of advanced solid tumors.”

- 6) **Page 10, under Study endpoints, also define the secondary endpoints of the study.**

Reply: Thanks for your comments. In fact, we defined the secondary endpoints on page 16 “The secondary endpoints included preliminary efficacy assessments based on ORR, progression-free survival (PFS), DoR, DCR, and clinical benefit rate (CBR), assessed per RECIST 1.1, as well as PK parameters for CYH33 and its metabolite I27.”

Hope we understood and answered your question correctly.

- 7) **Page 25, in the image for Fig 2, it is not possible to see the metastatic lesion (at least in the printed document. This should be checked again to make sure the lesion is visible to the reader.**

Reply: Thanks for your good suggestions. We revised the figure and circled the metastatic lesions as shown below:

Reviewer #3 - PI3K inhibitors, clinical trials (Remarks to the Author):

This manuscript by Wei and colleagues describes the first in man study of a novel PI3K α i inhibitor. This study shows acceptable pharmacokinetics for this compound and advises on a recommended Phase II dose based on tolerability.

The study confirms the benefit targeting the α isoform in patients with mutations and confirms activity in patients with *PIK3CA* mutations. Of note is the fact that this study was carried out in a Chinese populations.

The study is well carried out and it is exciting to see another PI3K α selective inhibitor in clinical study but there is nothing new here (apart from a novel compound) to warrant publication in nature communication.

Of note is the fact that the biomarker used is fasting glucose which is a distal biomarker on the pathway. From this, it is difficult to evaluate the real target engagement.

Multiple studies with PI3K inhibitors have used P-AKT in tumors and peripheral blood mononuclear cells, and this should have been done here as the PKPD relationships are easily established and known.

Overall, this is an interesting study, but this work should be submitted to a different more appropriate journal.

Reply: Thanks for recognizing the value of our study.

Regarding to PD markers, there have been multiple dimensions to comprehensively evaluate the target modulation, including direct target engagement biomarkers (such as pAKT) and indirect biomarkers (such as FBG, C-peptide, *etc*). Basing on the availability and actionability in clinical operation, we are keeping trying to collect the full package of these clinical biomarker data, and the pAKT assay derived from platelet-rich plasma has been underway in Phase Ib studies.

The importance of PI3K α selective inhibitor has been established in breast cancer. However, our study finds anti-tumor activity of CYH33 in several types of solid tumors, which suggests possibly broader indications and deserves further investigation.

Reviewers' Comments:

Reviewer #1:

Remarks to the Author:

My previous comments have been addressed by the authors. I have no further comments.

Reviewer #2:

Remarks to the Author:

Thank you for the revisions which have significantly improved the clarity of the manuscript.

Reviewer #3:

Remarks to the Author:

This manuscript is now suitable for publication